# VEditBench: Holistic Benchmark for Text-Guided Video Editing

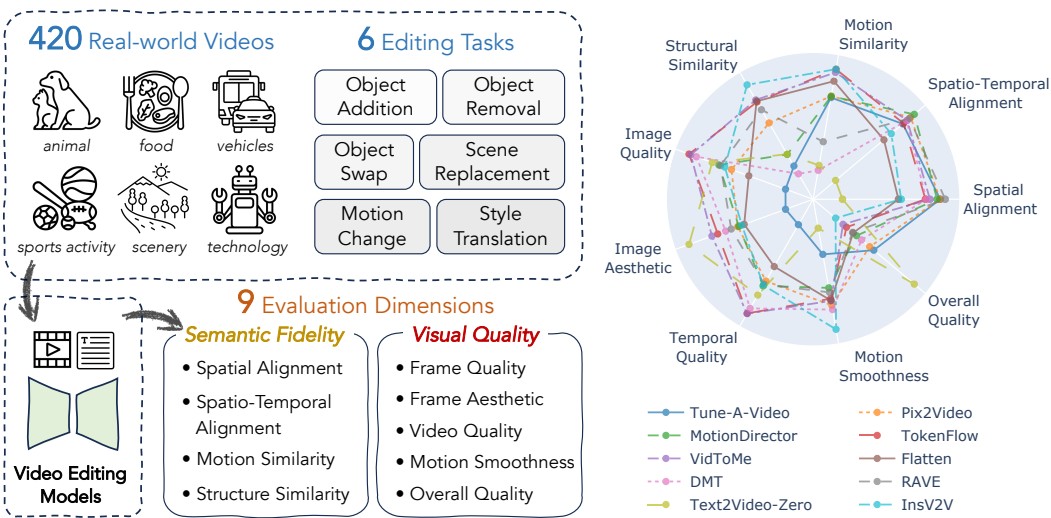

Figure 1: **Introducing VEditBench**, a holistic framework for the evaluation of Text-Guided Video Editing (TGVE) models. VEditBench features a diverse dataset of 420 real-world videos across six categories, along with six editing tasks driven by text prompts. We define nine distinct evaluation metrics to access the model's semantic fidelity and visual quality. Our evaluation of ten TGVE models using VEditBench provides a comprehensive analysis of their performance.

## ABSTRACT

Video editing usually requires substantial human expertise and effort. However, recent advances in generative models have democratized this process, enabling video edits to be made using simple textual instructions. Despite this progress, the absence of a standardized and comprehensive benchmark has made it difficult to compare different methods within a common framework. To address this gap, we introduce VEditBench, a comprehensive benchmark for text-guided video editing (TGVE). VEditBench offers several key features: (1) **420 real-world videos** spanning diverse categories and durations, including 300 short videos (2-4 seconds) and 120 longer videos (10-20 seconds); (2) **6 editing tasks** that capture a broad range of practical editing challenges: *object insertion*, *object removal*, *object swap*, *scene replacement*, *motion change*, and *style translation*; (3) **9 evaluation dimensions** to assess the semantic fidelity and visual quality of edits. We evaluate ten state-of-the-art video editing models using VEditBench, offering an in-depth analysis of their performance across metrics, tasks, and models. We hope VEditBench will provide valuable insights to the community and serve as the standard benchmark for TGVE models following its open-sourcing.

## 1 INTRODUCTION

The recent explosion of generative AI models has revolutionized content creation, with video editing emerging as a critical application in this rapidly evolving landscape. Millions of videos are produced

Table 1: **Existing benchmarks for text-guided video editing.** Many studies rely on private and non-standardized benchmarks, while existing open-source TGVE benchmarks are inadequate in terms of data scale and diversity.

| Paper | #Videos | Video Duration | Video Source | #Edit Prompts | Open-source |
|---|---|---|---|---|---|
| Tune-A-Video (Wu et al., 2023c) | 42 | 1-4s | DAVIS | 140 | ✗ |
| Dreamix (Molad et al., 2023) | 29 | - | YouTube-8M | 127 | ✗ |
| Gen-1 (Esser et al., 2023) | - | - | DAVIS | 35 | ✗ |
| Rerender A Video (Yang et al., 2023) | 8 | - | Pexels, Pixabay | - | ✗ |
| TokenFlow (Geyer et al., 2023) | 61 | 40-200 frames | DAVIS, Internet | 61 | ✗ |
| FlowVid (Liang et al., 2023) | 25 | 1-4s | DAVIS | 115 | ✗ |
| STDF (Yatim et al., 2023) | 21 | - | - | 54 | ✗ |
| Fairy (Wu et al., 2023a) | 50 | - | ShutterStock | 1000 | ✗ |
| RAVE (Wu et al., 2023a) | 186 | 8 / 36 / 90 frames | Pexel, Pixaba, DAVIS, Internet | 186 | ✗ |
| TGVE-2023 (Wu et al., 2023d) | 76 | 32 / 128 frames | DAVIS, YouTube, Videvo | 304 | ✓ |
| BalanceCC (Feng et al., 2024) | 100 | 2-20s | - | 400 | ✓ |
| V2VBench (Sun et al., 2024b) | 50 | 2-200s | Internet | 150 | ✓ |
| **VEditBench (Ours)** | 420 | 2-4s / 10-40s | YouTube, Videvo | 2520 | ✓ |

daily, and AI-driven tools are increasingly sought after to streamline and enhance the editing process. However, evaluating and comparing these text-guided video editing (TGVE) models presents a significant challenge due to the lack of a standardized and comprehensive benchmark.

Existing efforts to evaluate TGVE models suffer from several limitations. Many studies rely on small, private datasets that lack diversity and fail to reflect real-world editing scenarios (Wu et al., 2023c; Molad et al., 2023; Esser et al., 2023). This reliance on non-standardized and inaccessible data hinders fair and open comparisons between different approaches. While recent works like LOVEU-TGVE-2023 (Wu et al., 2023d), BalanceCC (Feng et al., 2024), and V2VBench (Sun et al., 2024b) have introduced open-source benchmarks, they remain limited in terms of data scale, prompt diversity, and the range of editing tasks they cover. These limitations underscore the urgent need for a more robust and comprehensive benchmark that can effectively assess the capabilities of TGVE models.

To address this gap, we introduce `VEditBench`, a comprehensive benchmark specifically designed for evaluating text-guided video editing. `VEditBench` provides a unified framework for assessing the performance of diverse video editing models across a wide range of real-world scenarios.

`VEditBench` distinguishes itself through three key advancements:

- **Diverse and Extensive Video Collection**: We curated a diverse collection of videos from YouTube and Videvo, spanning six categories: *Animals*, *Food*, *Scenery*, *Sports Activity*, *Technology*, and *Vehicles*. Recognizing the need for both short-form and long-form video editing, we include videos ranging from 2-4 seconds to more challenging 10-40 second clips, addressing a gap in existing benchmarks that primarily focus on short videos.

- **Expanded Scope of Editing Tasks**: `VEditBench` expands the scope of editing tasks beyond the limitations of previous benchmarks. Instead of focusing solely on foreground, background, and style modifications, we incorporate six diverse editing tasks reflective of real-world applications: *object insertion*, *object removal*, *object swap*, *scene replacement*, *motion change*, and *style translation*. This expanded task set allows for a more comprehensive evaluation of model capabilities across various editing scenarios.

- **Multi-Dimensional Evaluation Framework**: `VEditBench` addresses the challenge of evaluating video edits by employing a multi-dimensional evaluation framework. This framework encompasses both Semantic Fidelity (*i.e.*, *how accurately the edited video adheres to the user's command*) and Visual Quality (*i.e.*, *the overall visual appeal of the edited video, independent of the edit itself*). Within each perspective, we define specific sub-dimensions to enable a more fine-grained and insightful analysis of model performance.

To demonstrate the utility of `VEditBench`, we evaluate ten state-of-the-art video editing models, offering an in-depth analysis of their performance across different dimensions, tasks, and model architectures. This analysis provides valuable insights into the current state of TGVE and highlights areas for future research and development. `VEditBench` will be made fully open-source to foster further advancements in the field.

## 2 RELATED WORK

### 2.1 TEXT-GUIDED VIDEO EDITING (TGVE) MODELS.

TGVE aims to modify the visual content of a video based on textual prompts while preserving its inherent characteristics. Pioneer Tune-A-Video (Wu et al., 2023c) inflates the image diffusion models by incorporating cross-frame attention and fine-tuning on source videos to implicitly learn and transfer motion. While demonstrating versatility across various editing tasks, Tune-A-Video suffers from limitations in temporal consistency.

Subsequent works focus on extracting various correspondences from the source video to enhance temporal consistency. Methods like FateZero (Qi et al., 2023), Video-P2P (Liu et al., 2023a), and VidToMe (Li et al., 2024) extract cross- and self-attention features from the source video to guide spatial layout and maintain coherence across frames. Others, such as Rerender A Video (Yang et al., 2023), TokenFlow (Geyer et al., 2023), and Flatten (Cong et al., 2023b), focus on extracting and aligning optical flows to improve the consistency of editing results. Meanwhile, Text2Video-Zero (Khachatryan et al., 2023) and RAVE (Kara et al., 2024) utilize spatial conditioning techniques from ControlNet (Zhang & Agrawala, 2023) to guide the editing process. Instruct Video-to-Video (Cheng et al., 2023) explores instruction-guided video editing and investigates sampling techniques for consistent long video generation.

More recently, with the emergence of advanced text-to-video (T2V) foundation models, researchers have begun leveraging these models for improved temporal consistency in TGVE. MotionDirector fine-tune T2V diffusion models with disentangled spatial and temporal LoRA modules for motion customization. Diffusion Motion Transfer (DMT) (Yatim et al., 2024) employs a space-time feature loss derived directly from the model to preserve overall motion during editing.

Despite these advancements, the field of TGVE still lacks a standardized benchmark for evaluating and comparing different models. To address this critical gap, we introduce VEditBench, an open and comprehensive benchmark designed to facilitate the standardized evaluation of TGVE models

### 2.2 BENCHMARKS FOR VIDEO GENERATIVE MODELS.

Early efforts rely on datasets like UCF-101 (Soomro et al., 2012), MSR-VTT (Xu et al., 2016), and Kinetics (Carreira & Zisserman, 2017b; Carreira et al., 2018), which offer limited diversity. Make-A-Video (Singer et al., 2023) evaluates on 300 text prompts across five common categories, while FETV (Liu et al., 2023c) introduces fine-grained category labels and temporal dimensions for a more in-depth assessment. EvalCrafter (Liu et al., 2023b) expands the scope with 700 real-world prompts, and VBench (Huang et al., 2024) designs a compact yet representative prompt suite across various evaluation dimensions and content categories. T2V-CompBench (Sun et al., 2024a) focuses specifically on compositional text-to-video generation with 700 prompts spanning seven compositional categories.

While these works advance the evaluation of text-to-video generation, video editing benchmarks remain limited. LOVEU-TGVE-2023 (Wu et al., 2023d) introduces the first benchmark for text-guided video editing, featuring 76 videos and 304 edit prompts across four edit types. Similarly, BalanceCC (Feng et al., 2024) includes 100 videos, each paired with four edit prompts. However, both benchmarks lack sufficient video variety and task diversity.

To address these limitations, we propose VEditBench, a comprehensive benchmark comprising 420 diverse real-world videos, each annotated with six fine-grained edit tasks. Importantly, VEditBench includes 120 long videos (10-40 seconds), addressing the under-explored challenge of long video editing.

### 2.3 EVALUATION METRICS FOR VIDEO GENERATIVE MODELS.

Image-level metrics assess the quality of individual frames in generated videos. Common metrics include Inception Score (IS) (Barratt & Sharma, 2018) for image quality and diversity, Fréchet Inception Distance (FID) (Parmar et al., 2022) for similarity to real images, and CLIP Score (Radford et al., 2021) for alignment between images and text descriptions.

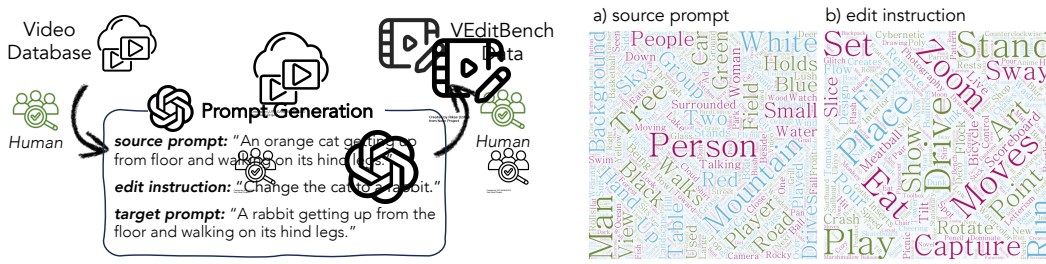

Figure 2: **VEditBench data curation pipeline that involves both machine and human.**

Figure 3: **Visualization of word distribution in source and edit prompt.**

Video metrics prioritize temporal aspects. Fréchet Video Distance (FVD) (Unterthiner et al., 2019) uses features from I3D (Carreira & Zisserman, 2017a) to compute the distance between generated and real video distributions, but can be biased towards frame quality over motion realism. To address this, Content-Debiased FVD (Ge et al., 2024) utilizes features from large-scale unsupervised models. Frame Consistency CLIP Score (Radford et al., 2021) measures the consistency of edited videos by comparing CLIP embeddings across frames.

Recent work has introduced dedicated metrics for T2V evaluation, such as T2VScore (Wu et al., 2024), VBench (Huang et al., 2024) and EvalCrafter (Liu et al., 2023b). Building upon prior research, we incorporate established metrics and introduce new ones tailored for video editing tasks, including scores for motion and sturctural similarity between source and edited video.

## 3 BENCHMARK CURATION

### 3.1 COLLECTION OF VIDEOS.

We aim to curate a diverse benchmark for real-world video editing applications. We consider six categories from everyday life: *Animal*, *Food*, *Scenery*, *Sports Activity*, *Technology*, and *Vehicle*. We search two large-scale video databases: YouTube[1] and Videvo[2]. YouTube serves as one of the largest video repositories, featuring diverse user-generated content, while Videvo offers high-quality stock videos shot by professionals.

To diversify the video content, we first ask GPT-4o to provide distinct keywords for each category and use these keywords to search within the Panda-70M dataset (Chen et al., 2024) and YouTube. To ensure data quality, we manually check each video and filter out those of low quality (*e.g.*, blurry, shaking, ghosting). We obtain the video captions using GPT-4o. Since the captions generated by large multimodal models may exhibit issues such as missing objects or hallucinations of non-existent objects (Bai et al., 2024), we also dedicate manual effort to reviewing and revising the captions, ensuring that the key pixels are accurately described.

Mainstream TGVE models typically focus on short video editing, handling clips of 2 to 4 seconds (24-30fps) in length (usually under 100 frames). To support this, we collect 300 short videos within this range. Additionally, we explore a more challenging task: editing longer videos of 10 to 40 seconds (24-30fps). This task presents greater difficulty, as it requires the model to maintain long-range consistency in video content (*e.g.*, subject and style) across transitions. Solving this challenge will make TGVE models more practical and applicable to real-world scenarios, such as the film production.

Finally, we curate a collection of 420 videos, comprising 300 short videos and 120 long videos, all at a resolution of 720×1280. These videos are balanced across and diversified within six categories.

---

[1]https://www.youtube.com/
[2]https://www.videvo.net/

Figure 4: **Illustration of six video editing tasks in `VEditBench`.**

## 3.2 DESIGN OF VIDEO EDITING TASKS.

The existing literature on video editing primarily addresses changes to the subject, background, and style. In this work, we explore broader applications of video editing and define six distinct video editing tasks as follows:

- *Object Addition*: add new objects to the video (*e.g.*, "add a string toy near the cat")
- *Object Removal*: remove existing objects from the video (*e.g.*, "remove the cat")
- *Object Swap*: replace the object while maintaining its motion (*e.g.*, "change the cat to a rabbit")
- *Scene Replacement*: change the location (*e.g.*, "place the cat in a grassy field")
- *Motion Change*: modify the object's or camera's motion (*e.g.*, "tilt the camera downwards")
- *Style Translation*: apply a specific style (*e.g.*, "make it in Van Gogh style")

Each of these tasks serves a distinct purpose in examining the capability of TGVE models. We illustrate each editing task in Figure 13.

We task GPT-4o with the above descriptions to generate diverse edit prompts. Specifically, we feed sampled video frames in a grid along with the video caption to GPT-4o, which then returns the corresponding edit instructions and target prompts for each task (see Figure 2). Still, we manually review all the machine-generated prompts with necessary modifications to ensure accuracy. In Figure 3, we visualize the word distribution in our source and edit prompt set. More details about edit prompt generation can be found in the supplementary material.

## 4 EVALUATION METRICS

We assess the performance of TGVE models from two primary perspectives: 1) **Semantic Fidelity** – *Does the edited video adhere to the user's command?*, which evaluates whether the output video accurately follows the guidance from input video and edit prompt. 2) **Video Quality** – *Regarding of the editing instructions, is the generated video visually appealing?*, which focuses on the overall visual quality of the resulting video, independent of the applied edits. For each of these perspectives, we further define several sub-dimensions to enable a more fine-grained evaluation.

### 4.1 SEMANTIC FIDELITY

A successfully edited video should accurately follow: 1) the *explicit* instructions provided by users (*i.e.*, user prompt); 2) the *implicit* consistency with the source video (*e.g.*, motion, structure, that are not intended for editing). To this end, we break down Semantic Fidelity into two distinct aspects, *Text Alignment* and *Video Alignment*, where the former focuses on the faithfulness with the target prompt, and the latter considers the coherence with the source video.

**[Text] Spatial Alignment.** The CLIP model (Radford et al., 2021) trained on massive text-image pairs is capable of encoding meaningful embeddings for both modalities in a shared latent space.

It is widely used to measure the similarity between visual and textual data. We compute the CLIP feature similarity between the generated frames and their corresponding target prompts.

**[Text] Spatio-Temporal Alignment.** In addition to spatial content, videos display temporal dynamics such as object movement and camera motions. Huang et al. (2024) demonstrate the effectiveness of using a video CLIP model, *i.e.*, ViCLIP (Wang et al., 2023b), to evaluate text-video alignment for text-to-video generation. We measure the Spatio-Temporal Text Alignment by calculating the feature similarity between the ViCLIP embeddings of edited video and target prompt.

**[Video] Structural Similarity.** In video editing, it is essential to preserve the integrity of the original content. We compute the Structural Similarity Index Measure (SSIM) (Wang et al., 2004) between source and corresponding target frames. SSIM compares the structural features of the source and target videos, and helps identify any significant alterations that may compromise the original message.

**[Video] Motion Similarity.** The goal is to quantify how much the motion dynamics change between a source video and a target video. We first estimate a set of point trajectories $\mathbf{T} = \{(\mathbf{p}_i, \mathbf{v}_i)\}_{i=1}^{N}$, using the off-the-shelf CoTracker (Karaev et al., 2023). Here $\mathbf{p}_i$ and $\mathbf{v}_i$ represents the position and motion vector at $i$-th trajectory, with $N$ being the total number of trajectories extracted in video. We denote the trajectory sets for source video and target video as $\mathbf{T}^A$ and $\mathbf{T}^B$.

To compare these trajectory sets, we define a combined cost matrix for the $i$-th trajectory from video $A$ and the $j$-th trajectory from video $B$. The matrix considers both positional and directional differences between the trajectories:

$$C(i,j) = \alpha \cdot \underbrace{\frac{\|\mathbf{p}_i^A - \mathbf{p}_j^B\|_2}{D_{\max}}}_{\text{Positional Cost}} + (1-\alpha) \cdot \underbrace{\left(1 - \frac{\mathbf{v}_i^A \cdot \mathbf{v}_j^B}{\|\mathbf{v}_i^A\|_2 \|\mathbf{v}_j^B\|_2 + \epsilon}\right)}_{\text{Directional Cost}},$$

where $D_{\max}$ is the maximum observed distance used for normalization, $\alpha \in [0,1]$ is a weighting parameter balancing positional and directional terms, and $\epsilon$ is a small constant to avoid zero division.

We employ the Hungarian algorithm (Kuhn, 1955) to find the optimal assignment of trajectories between the two videos, minimizing the total cost: $\min_\sigma \sum_i C_{i\sigma(i)}$, where $\sigma(i)$ maps trajectory $i$ in video $A$ to a corresponding trajectory in video $B$. Finally, we compute the motion similarity score between the two videos as: $S_{\text{MotionSim}} = 1 - \frac{1}{N}\sum_i C_{i\sigma(i)}$.

This score indicates how closely the motion patterns align between the videos. A higher score reflects greater similarity. Empirically, we set equal weights for the positional and directional terms, i.e., $\alpha = 0.5$, to balance their contributions.

## 4.2 Visual Quality

Video can be seen as a sequence of images with consistent temporal dynamics. We evaluate the visual quality of a video from three perspectives: 1) **Spatial Quality**, which analyzes the video as individual frames, independent of temporal dynamics, by calculating the average image score across the frames; 2) **Temporal Quality**, which focuses solely on the temporal dimension, assessing the consistency of the video over time; 3) **Spatio-Temporal Quality**, which considers the video as a whole, integrating both spatial and temporal elements.

**[Spatial] Image Quality.** Image quality focuses on the impact of distortions and other visual imperfections in images on human perception. Recently, Wu et al. (2023b) introduce Q-Align, an advanced approach that trains large multimodal models to perform visual scoring. Q-Align demonstrates a significant leap in image quality assessment, image aesthetic assessment and video quality assessment – not only achieving state-of-the-art performance but also enhancing out-of-distribution generalization capabilities. We adopt Q-Align as the method for image quality scoring.

**[Spatial] Image Aesthetic.** Image aesthetic measures the visual appeal and beauty of an image. We evaluate it using the Q-Align's image aesthetic scorer trained on AVA dataset (Gu et al., 2018).

**[Temporal] Motion Smoothness.** Motion smoothness refers to the continuity of movement in visual content, often measured by the absence of noticeable jitter, stuttering, or abrupt transitions between frames. We follow VBench (Huang et al., 2024) to use the motion priors from the video frame interpolation model (Li et al., 2023) to assess the smoothness of motion in edited videos.

**[Temporal] Temporal Quality.** Fréchet Video Distance (FVD) is a widely used metric for assessing the temporal quality of generated videos. It measures the similarity between the distributions of real and generated videos by comparing the feature representations extracted from a pre-trained neural network. However, Ge et al. (2024) found that FVD tends to prioritize per-frame quality over temporal consistency. They attribute this bias to the features derived from a supervised video classifier trained on a content-biased dataset. To address this issue, they suggest using features from large-scale unsupervised models, which can help mitigate the bias. We employ their implementation of Content-Debbiased FVD[3], calculated using VideoMAE-v2 (Wang et al., 2023a) features, to evaluate temporal quality.

**[Spatio-Temporal] Video Quality.** This dimension takes into account both spatial and temporal factors, offering a comprehensive understanding of a video's performance. Q-Align (Wu et al., 2023b) utilizes a language decoder to assemble videos as sequences of frames, so as to unify video quality assessment with image quality/aesthetic assessment under one structure. It also marks state-of-the-art in video quality assessment; therefore, we utilize it as video quality scorer.

## 5 EXPERIMENTS

**Evaluated Models.** We evaluate ten TGVE models on `VEditBench`, including Tune-A-Video (Wu et al., 2023c), MotionDirector (Zhao et al., 2023), VidToMe (Li et al., 2024), Pix2Video (Ceylan et al., 2023), TokenFlow (Geyer et al., 2023), Flatten (Cong et al., 2023a), Diffusion Motion Transfer (DMT) (Yatim et al., 2024), RAVE (Kara et al., 2024), Text2Video-Zero (Khachatryan et al., 2023), and Instruct Video-to-Video (InsV2V) (Cheng et al., 2023). Among them, Text2Video-Zero and InsV2V accept editing instructions as input, whereas the others rely on a target prompt.

**Settings.** To account for the varying capabilities of TGVE models in handling different video lengths, we partition `VEditBench` into two subsets: `VEditBench-Short` and `VEditBench-Long`, designed for evaluating short and long video editing, respectively. `VEditBench-Short` includes all ten models outlined above, enabling a comprehensive comparison of their performance on short videos. However, since some models are not optimized for long video editing, `VEditBench-Long` focuses on evaluating four models specifically designed or adapted: Pix2Video, Text2Video-Zero, VidToMe, and InsV2V.

**Results.** To comprehensively assess the performance of different TGVE models on `VEditBench`, we conduct both quantitative and qualitative analyses. Our quantitative evaluation leverages a diverse set of metrics designed to measure various aspects of video quality and fidelity to the editing instructions (Table 3, Figure 5). Complementing these quantitative measures, we also perform a qualitative analysis to provide a more nuanced understanding of the strengths and weaknesses of each model (Figure 6). This involves visual inspection of the edited videos and a comparative analysis of their performance across different editing tasks and video categories.

## 6 INSIGHTS AND DISCUSSIONS

**No Single Model Dominates Across All Dimensions.** As shown in Table 3, no single TGVE method consistently excels across all evaluation dimensions. Each model demonstrates strengths in specific areas while exhibiting weaknesses in others, highlighting the diverse approaches and trade-offs within the field. For instance, while RAVE achieves strong performance in Spatial and Spatio-Temporal Alignment, it lags in terms of visual quality, as evidenced by its lower scores in Image Quality, Image Aesthetics, and Video Quality. The irregular shapes of the radar charts (Figure 5) also indicate that there are often trade-offs between different evaluation metrics. A model might

---

[3]https://github.com/songweige/content-debiased-fvd

Table 2: **Results per dimension on `VEditBench-Short`.** This table compares the performance of ten TGVE models across nine dimensions. The best and second-best are **bold-faced** and underlined. Efficiency measures TGVE models' runtime (seconds per frame, SPF) and GPU memory usage (Mem) on an NVIDIA A100 GPU. †T2I-based method, ‡T2V-based method.

| | Spatial Alignment | SpatioTemp Alignment | Motion Sim. | Structural Sim. | Image Quality | Image Aesthetic | Video Quality | Motion Smooth. | Temporal Quality | Efficiency (SPF / Mem) |
|---|---|---|---|---|---|---|---|---|---|---|
| Tune-A-Video† | 26.550 | 0.239 | 0.887 | 0.447 | 0.399 | 0.233 | 0.467 | 0.942 | 401.023 | 30.1s / 16GB |
| Pix2Video† | 26.543 | 0.248 | 0.889 | 0.604 | 0.592 | 0.375 | 0.665 | 0.971 | 367.610 | 11.8s / 27GB |
| MotionDirector‡ | 26.393 | **0.252** | 0.889 | 0.489 | 0.636 | 0.372 | 0.682 | 0.961 | 262.489 | 12.5s / 20GB |
| TokenFlow† | 25.806 | 0.240 | **0.925** | 0.681 | **0.743** | 0.435 | 0.778 | 0.967 | 181.586 | 6.4s / 7GB |
| VidToMe† | 26.033 | 0.244 | 0.920 | 0.688 | 0.736 | 0.452 | **0.779** | 0.968 | 153.368 | 5.3s / 6GB |
| Flatten† | 24.448 | 0.217 | 0.909 | 0.683 | 0.530 | 0.356 | 0.614 | 0.968 | 235.446 | 7.5s / 13GB |
| DMT‡ | 25.849 | 0.243 | 0.791 | 0.418 | 0.716 | 0.411 | 0.761 | 0.973 | 302.740 | 20.3s / 40GB |
| RAVE† | **26.801** | 0.246 | 0.829 | 0.652 | 0.631 | 0.395 | 0.676 | 0.964 | 230.579 | 3.2s / 26GB |
| Text2Video-Zero† | 21.631 | 0.162 | 0.798 | 0.490 | 0.660 | **0.520** | 0.714 | 0.927 | 725.644 | 3.1s / 23GB |
| InsV2V‡ | 24.586 | 0.226 | **0.925** | **0.743** | 0.615 | 0.363 | 0.680 | **0.984** | **94.294** | 2.6s / 14GB |

Table 3: **Results per dimension on `VEditBench-Long`.** This table compares the performance of ten TGVE models across nine dimensions.

| | Spatial Alignment | SpatioTemp Alignment | Motion Sim. | Structural Sim. | Image Quality | Image Aesthetic | Video Quality | Motion Smooth. | Temporal Quality |
|---|---|---|---|---|---|---|---|---|---|
| Pix2Video | 26.741 | 0.243 | 0.841 | 0.597 | 0.609 | 0.365 | 0.684 | 0.972 | 505.415 |
| VidToMe | 26.371 | 0.239 | 0.876 | 0.675 | 0.723 | 0.430 | 0.791 | 0.971 | 269.596 |
| Text2Video-Zero | 22.767 | 0.174 | 0.771 | 0.477 | 0.502 | 0.753 | 0.714 | 0.932 | 869.299 |
| InsV2V | 25.551 | 0.226 | 0.906 | 0.740 | 0.689 | 0.383 | 0.742 | 0.987 | 140.232 |

score high on image quality but lower on motion smoothness, suggesting that optimizing for one metric can sometimes come at the expense of another.

Notably, TokenFlow and VidToMe emerge as more well-rounded models, achieving high performance in visual quality while maintaining strong semantic fidelity scores. These findings underscore the importance of a comprehensive benchmark like `VEditBench` to provide a nuanced understanding of model performance and guide future research towards more robust and versatile TGVE methods.

**Model Performance Varies Across Tasks.** The charts in Figure 5 clearly show that a model's performance can vary significantly depending on the specific editing task. For instance, some models excel at object swap but struggle with motion change. This highlights the importance of evaluating models across a diverse range of tasks to understand their strengths and weaknesses.

**Semantic Fidelity *vs*. Visual Quality** Our analysis reveals an interesting tension between semantic fidelity and visual quality in TGVE models. While some models excel at accurately adhering to the editing instructions (high semantic fidelity), they may sometimes produce outputs with noticeable visual artifacts or inconsistencies (lower visual quality). Conversely, other models prioritize generating visually appealing results but may struggle to precisely fulfill the user's intent. This trade-off highlights a key challenge in TGVE: achieving a balance between accurately interpreting and executing editing instructions while maintaining high visual quality in the output. Future research could explore novel approaches to optimize both aspects simultaneously, potentially through improved training strategies or more sophisticated evaluation metrics that explicitly consider the interplay between semantic fidelity and visual quality.

**Challenges in Long Video Editing.** Evaluating models on `VEditBench-Long` reveals unique challenges associated with editing longer videos. Maintaining temporal consistency and coherence over extended durations proves to be a significant hurdle for most models. Edited outputs exhibit increased occurrences of flickering, temporal artifacts, and deviations from the original video's narrative flow. These challenges stem from the increased complexity of modeling long-range dependencies and the potential for errors to accumulate over time. Furthermore, computational constraints become more prominent when processing longer videos, which can limit the effectiveness of certain techniques. These findings highlight the need for further research focused on developing specialized architectures and training strategies tailored to the specific challenges of long video editing.

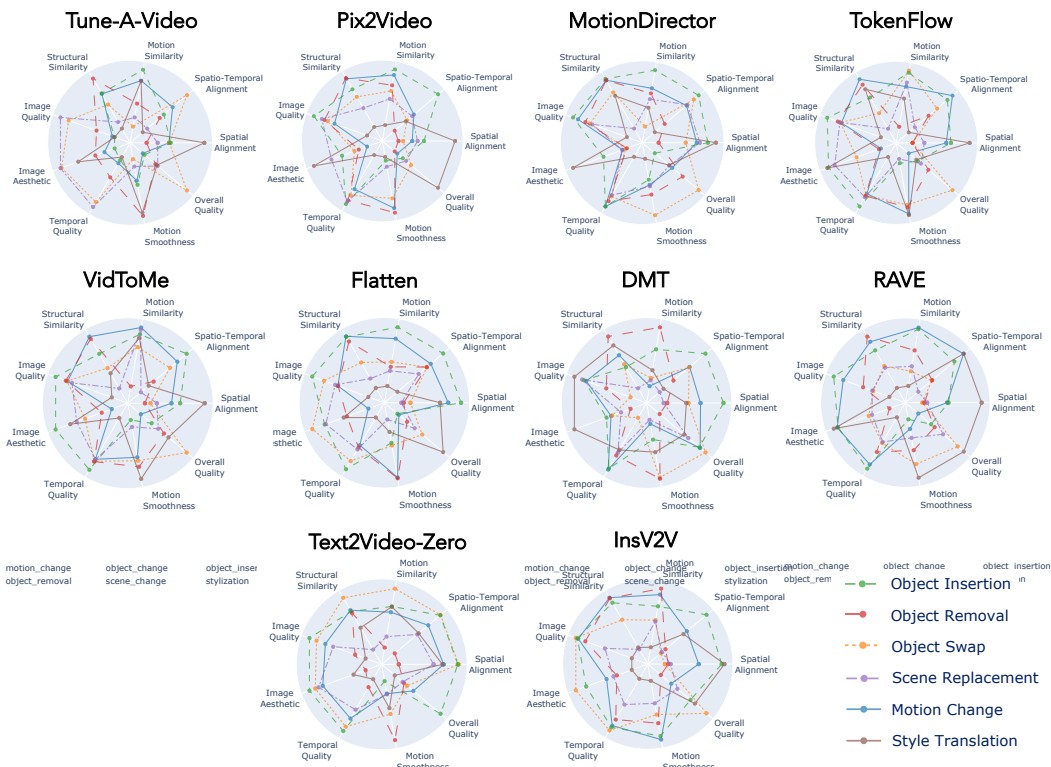

Figure 5: **Results per model on `VEditBench-Short`.** We visualize each model's performance across six editing tasks and nine evaluation dimensions. The radar charts reveal that model performance varies significantly across tasks, highlighting the importance of comprehensive evaluation across diverse editing scenarios.

## 7 CONCLUSION

In this paper, we introduced `VEditBench`, a comprehensive benchmark designed to standardize and advance the evaluation of text-guided video editing models. `VEditBench` addresses key limitations of existing benchmarks by providing a diverse collection of real-world videos, a wider range of editing tasks, and a multi-dimensional evaluation framework encompassing both semantic fidelity and visual quality. By evaluating ten state-of-the-art TGVE models on `VEditBench`, we offer insights into their capabilities and highlight areas for future improvement. We believe that the open-source release of `VEditBench` will serve as a valuable resource for the research community, fostering further progress in this rapidly evolving field.

**Limitation and Future Work.** The benchmark currently focuses on single-shot edits based on a single textual instruction. Future work could explore more complex editing scenarios involving multi-step edits or the composition of multiple instructions. We also plan to benchmark more TGVE models using our `VEditBench` in the future.

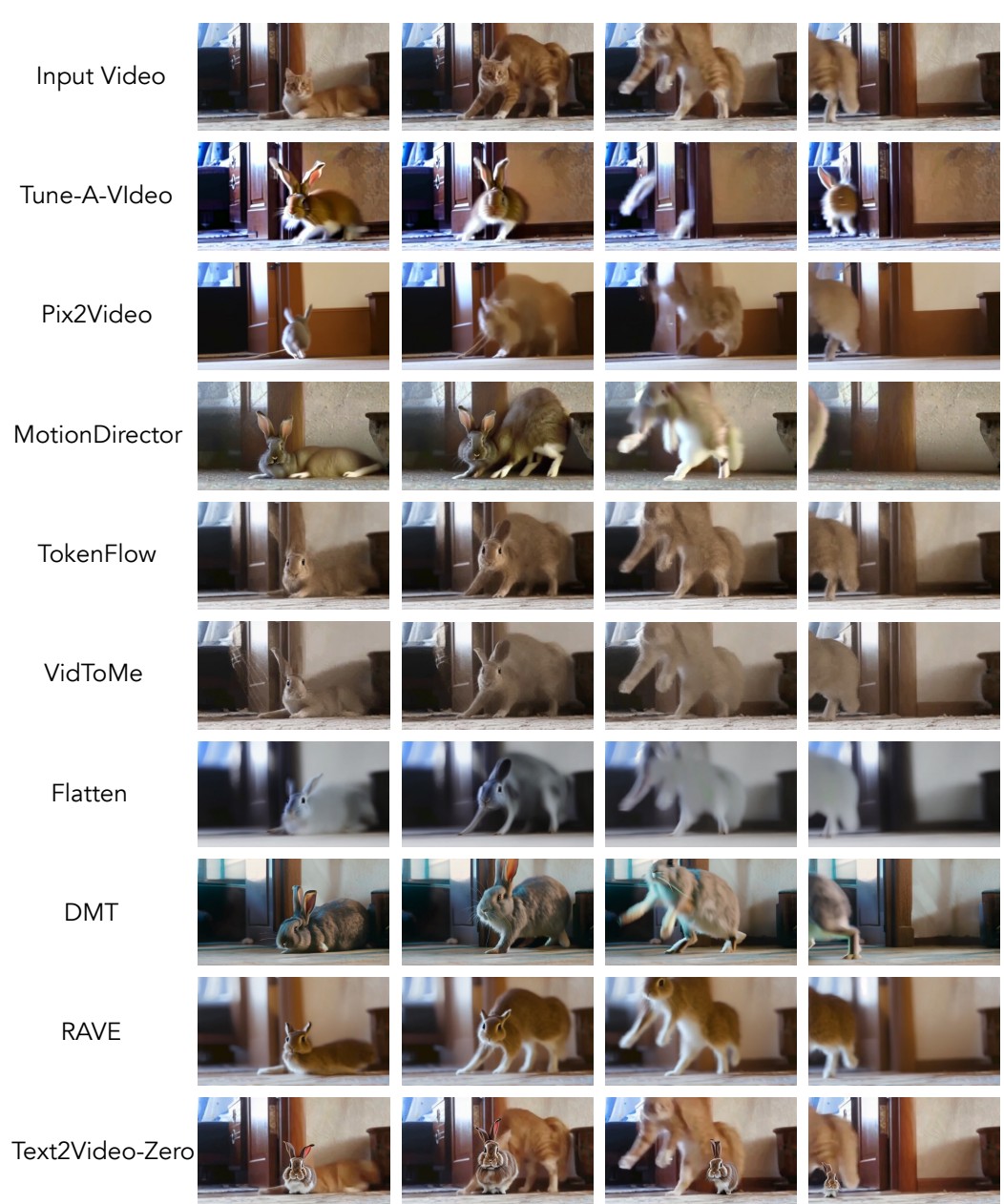

Figure 6: **Example of "changing the cat to a rabbit" in `VEditBench`.**

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
