# OpenReview forum: "VEditBench: Holistic Benchmark for Text-Guided Video Editing"
_ICLR.cc/2025/Conference — Submitted to ICLR 2025_

### Official Review · Reviewer_DDax · 2024-10-30

**Soundness:** 2
**Presentation:** 3
**Contribution:** 3
**Rating:** 5
**Confidence:** 2

**Summary:**

This paper presents VEditBench, a new benchmark for text-guided video editing (TGVE). VEditBench enjoys several key features, including (1) 420 real-world videos spanning diverse categories and durations, consisting of 300 short videos (2-4 seconds) and 120 longer videos (10-20 seconds); (2) defining six editing tasks that capture a broad range of practical editing challenges like object insertion, object removal, object swap, and style translation; (3) suggesting nine evaluation dimensions to assess the semantic fidelity and visual quality of edits. Furthermore, the authors evaluate ten state-of-the-art video editing models using VEditBench, and conduct an in-depth analysis of their performance across metrics, tasks, and models.

**Strengths:**

The main strength of this paper is that the authors introduce a new benchmark for text-guided video editing by collecting 420 short and long videos, which includes more videos than previous benchmarks. The authors also suggest some metrics for evaluation and some editing tasks for experiments. This paper is also well-written, making the main text easy to follow and easy to read. The authors also conduct numerous expeirments of recent state-of-the-art methods using VEditBench, offering a comprehensive comparison and experiments.

**Weaknesses:**

As a benchmark paper, I would expect that the authors provide open-source datasets or links to the dataset/website/codes, etc. Unfortunately, I cannot find them in this paper. I believe this is necessary and the authors should elaborate more on that.

Another concern is that this paper seems a bit incremental. Many of the metrics and tasks defined in this paper are actually come from prior works or have already being used in the community. Furthermore, the categories of the gathered videos are only restricted to six categories. The authors could consider further expanding them and including categories like clothes, plants, buildings, etc.

Furthermore, it seems that the tasks introduced by VEditBench is a bit simple. For example, as shown in Table 2 and Table 3, existing methods can achieve quite strong performance under many tasks and metrics in VEditBench.

**Questions:**

- what do you think are the most key and novel components of your benchmark compared to previous works?
- why do you not include some long videos (say 200s) like V2VBench? Would the editing tasks and metrics still be reliable for long videos (e.g., 200s)?

---

> ### Author Response · Authors · 2024-11-28
> **Response to Reviewer DDax**
>
> **[Q1: Datasets]**
>
> We have included a project page showcasing a sample dataset in the updated supplementary material. For more details, please refer to Sec. A.3. We confirm that the full dataset, along with the benchmark evaluation code and detailed documentation, will be made publicly available upon acceptance.
>
>
> **[Q2: Metrics and tasks]**
>
> Thank you for your thoughtful feedback. We acknowledge that some of our metrics and tasks build on prior works, which is an intentional decision to ensure reliability and comparability with established research. Metrics such as Spatio-Temporal Alignment and Motion Smoothness have been widely validated in text-to-video generation (e.g., VBench) and provide a robust foundation for evaluating text-guided video editing (TGVE). Beyond this, we introduce novel metrics like Motion Similarity and Structural Similarity, specifically designed to address underexplored aspects of TGVE, offering a more comprehensive evaluation framework than previous benchmarks.
>
> Similarly, while tasks such as object swap, background replacement, and stylization are common among prior works, VEditBench expands the task scope by introducing tasks like object addition, object removal, and motion change. These additions reflect real-world video editing scenarios that are critical yet underexplored in existing benchmarks, making VEditBench uniquely positioned to address practical challenges in TGVE.
>
> Regarding video categories, we have included six representative categories (Animal, Food, Vehicle, Sports Activity, Scenery, and Technology) to ensure a balance between diversity and practicality. However, we agree that extending these categories to include additional domains like clothing, plants, and buildings would further enhance the benchmark’s versatility. We are actively considering these expansions for future iterations of VEditBench.
>
> Thank you again for highlighting these areas, and we are committed to continuously improving the benchmark to better serve the TGVE research community.
>
> **[Q3: More complex task]**
>
> In response, we have expanded the dataset to incorporate more complex tasks, such as compositional editing, where multiple elements within a scene—such as object, motion, and style—are modified simultaneously in a single prompt. Examples of these prompts can be found in the Sec. A.4 of supplementary material. We will provide more detailed analysis on compositional editing task in the final paper.
>
>
> **[Q4: Key and novel components]**
>
> We propose VEditBench, a comprehensive and standardized benchmark for evaluating text-guided video editing (TGVE) models. VEditBench bridges critical gaps in the evaluation of TGVE methods by:
> - Expanding data diversity: We introduce a dataset of 420 real-world videos that span diverse categories and a wide range of durations. This addresses a significant gap in existing TGVE benchmarks, which are often limited in scale and predominantly focus on short videos.
> - Broadening task scope: VEditBench expands the scope of editing tasks beyond the limitations of previous benchmarks. We incorporate six diverse editing tasks reflective of various editing scenarios.
> - Implementing multi-dimensional evaluation: VEditBench addresses the challenge of evaluating video edits by employing a multi-dimensional evaluation framework. It defines specific sub-dimensions to enable a more fine-grained and insightful analysis of model performance.
>
> Together, these features make VEditBench a powerful tool for advancing the development and assessment of TGVE models.
>
> **[Q5: Longer videos]**
>
> Thank you for raising this important question. Our current focus is on evaluating video lengths ranging from 10 to 40 seconds, as this represents a significant challenge for existing TGVE models. Notably, none of the current state-of-the-art methods claim the ability to handle videos as long as 200 seconds with satisfactory results. Our evaluations already reveal that many models struggle with maintaining temporal consistency and coherence even at the 10-40 second range.
>
> We agree that extending the benchmark to include longer videos, such as 200 seconds, would be a valuable direction for future work. However, this would require advancements in TGVE methods capable of handling such lengths reliably. As these capabilities emerge, we plan to adapt VEditBench to incorporate and evaluate longer videos while ensuring the robustness of the tasks and metrics.

---

### Official Review · Reviewer_4M8x · 2024-11-01

**Soundness:** 3
**Presentation:** 3
**Contribution:** 2
**Rating:** 5
**Confidence:** 4

**Summary:**

This paper introduces a new benchmark, VEdit Bench, for evaluating text-guided video editing methods. The benchmark comprises 420 videos, including both short and long formats, and covers six distinct editing tasks. It provides a standardized evaluation framework with nine metrics that assess performance in terms of **semantic fidelity** and **visual quality**.  Furthermore, it conducts a comprehensive evaluation of existing video editing methods, providing insights into their performance across various tasks and metrics.

**Strengths:**

1. The new VEdit benchmark includes a significantly larger dataset, with 420 videos compared to the previous LOVEU-TGVE23’s 76 videos.
2. The benchmark addresses long video editing (10-40 seconds), which is essential for real-world applications.
3. VEdit Bench introduces a novel motion similarity metric that considers both positional and directional costs, distinguishing it from the motion fidelity metric in DMT.
4. The VEdit benchmark provides a comprehensive evaluation of most existing video editing methods, primarily those based on Stable Diffusion T2I models, with the exception of DMT, which is based on a T2V model. It also offers new insights, such as the importance of balancing semantic fidelity and visual quality as a direction for future research.

**Weaknesses:**

1. **Metric Weakness 1 (Coarse Semantic Alignment)**: The metric for semantic fidelity focuses primarily on text alignment and video alignment. However, relying solely on CLIP’s text and visual embeddings may not fully capture text alignment, as CLIP’s global representation is often coarse. The authors could enhance this by incorporating object masks, such as calculating visual-text alignment only within the foreground object’s mask region.

2. **Metric Weakness 2 (Editing Accuracy)**: For video editing methods based on T2I models, an essential issue is the model’s ability to accurately follow the instruction prompt. Therefore, **editing accuracy** should be a key metric to evaluate whether edits align with the prompt. Additionally, some methods unintentionally alter the background or other video elements even when only the object should be edited. Accurately assessing whether edits are limited to the specified target is essential.

3. **Task Weakness 1 (Multi-Object Editing)**: The benchmark lacks tasks sensitive to multiple object editing, such as multi-object video editing. For instance, VBench [1] includes specific metrics for multi-object editing, which could provide a more nuanced assessment of the model’s performance in handling complex scenes with multiple subjects.

4. **Task Weakness 2 (Subject Consistency)**: The benchmark does not adequately address subject consistency as a key quality indicator. Calculating overall motion consistency without distinguishing between foreground and background may be inadequate. Similar to VBench[1], separately evaluating subject consistency and background consistency could offer a more fine-grained metric for realistic video editing quality.

5. **Task Weakness 3 (Inadequate for T2I Models)**: The object addition and removal tasks fall under inpainting or outpainting. Current T2I models may lack the capability to perform object addition in a time-consistent manner, meaning these tasks likely require video generation models with temporal priors.

6. **Task Weakness 4 (Multi-Attribute Editing)**: The object swap task may involve simultaneous scene replacement, making it a challenging task. Editing multiple subjects or instances, also known as **multi-attribute editing**, should be considered a distinct task to evaluate complex editing capabilities.

7. **Task Weakness 5 (T2I Limitations on Motion)**: Tasks like motion change and object addition are challenging for T2I-based editing methods due to the lack of video generative priors. T2I models excel in visual quality but often lack temporal consistency, whereas T2V models offer better temporal consistency but lower aesthetic quality. Balancing the evaluation for both T2I and T2V models’ editing abilities is an important consideration.

8. **Visualization Weakness (Lack of Qualitative Demos)**: A comprehensive benchmark should include qualitative visualizations, such as video or GIF demos. VBench, for instance, visually demonstrates what a high or low temporal consistency score looks like, helping readers intuitively understand the metric. VEdit Bench, however, focuses heavily on quantitative metrics, lacking qualitative demonstrations that could help readers better grasp how current metrics align with human perception and what quality aspects they evaluate.

[1] Huang, Z. et al., "VBench: Comprehensive Benchmark Suite for Video Generative Models," CVPR, 2024.

**Questions:**

please see weaknesses above

**Details Of Ethics Concerns:**

no ethics concerns

---

> ### Author Response · Authors · 2024-11-28
> **Response to Reviewer 4M8x (1/2)**
>
> **[Q1: Metric Weakness 1 (Coarse Semantic Alignment)]**
>
> We appreciate the reviewer’s insightful comment and agree that incorporating object masks could provide a more fine-grained evaluation of text alignment and related dimensions. However, obtaining accurate masks for videos poses significant challenges. Manual annotation of masks is labor-intensive and requires substantial human effort to ensure precision. While automatic segmentation tools (e.g., SAM) exist, they often introduce errors, especially when handling small or partially occluded objects or low-quality synthetic videos. Please refer to Sec. A.7 of supplementary material for detailed analysis.
>
> These inaccuracies can compromise the robustness of the evaluation metrics, potentially introducing bias or noise into the results. We acknowledge the potential value of this approach and view it as an important direction for future research.
>
> **[Q2: Metric Weakness 2 (Editing Accuracy)]**
>
> We appreciate the reviewer’s emphasis on the critical aspect of editing accuracy, particularly ensuring edits align with the prompt without unintended modifications to other video elements. VEditBench already addresses this through metrics such as Spatial Alignment, Spatio-Temporal Alignment, Structural Similarity, and Motion Similarity, which evaluate adherence to prompts and the preservation of source content. However, we agree that a more targeted assessment is warranted.
>
> As mentioned in response to the previous question, segmentation-based analysis offers a promising approach for isolating and evaluating unintended changes in non-target regions. However, obtaining accurate object masks for videos during the testing phase poses significant challenges. Manual annotation is labor-intensive, while automatic segmentation tools often introduce errors, as illustrated in Fig. 12. These issues could undermine the robustness of the evaluation metrics.
>
> We view this as an important avenue for future research and plan to explore efficient methods for integrating high-quality object masks into subsequent iterations of VEditBench to further improve evaluation accuracy and reliability.
>
> **[Q3: Task Weakness 1 (Multi-Object Editing)]**
>
> Thank you for the suggestion. Our benchmark already includes multi-object editing scenarios, where multiple objects within a scene are modified simultaneously. For example, in the 17th example on the project page, a group of people is transformed into a group of robots. We agree with the reviewer that multi-object editing could be elevated as a standalone task in video editing, and we plan to further explore and refine this task in future iterations of VEditBench.
>
>
> **[Q4: Task Weakness 2 (Subject Consistency)]**
>
> VBench uses DINO feature similarity to evaluate subject consistency and CLIP scores to assess background consistency. However, these methods still rely on global features and do not explicitly separate the foreground and background. We agree that fine-grained metrics, capable of isolating and evaluating these components independently, are crucial for a more realistic assessment of video editing quality.
>
> We recognize the potential of mask-based evaluation, which would allow precise differentiation between subjects and backgrounds. As noted earlier, challenges such as obtaining accurate masks—whether through manual annotation or automated tools—must be resolved to ensure reliable evaluation. We see this as a critical area for future research and plan to incorporate high-quality, mask-based metrics into future iterations of VEditBench to enhance the evaluation of subject consistency and other dimensions of video quality.

---

> > ### Author Response · Authors · 2024-11-28
> > **Response to Reviewer 4M8x (2/2)**
> >
> > **[Q5: Task Weakness 3 (Inadequate for T2I Models)]**
> >
> > We appreciate the reviewer’s feedback and would like to clarify that the perceived limitation pertains to certain T2I models rather than our benchmark. Object addition and removal tasks are indeed challenging for T2I models, particularly in maintaining temporal consistency. However, advancements in temporal adaptations for T2I models—such as temporal attention mechanisms and optical flow techniques—demonstrate that achieving time-consistent edits is feasible without relying solely on full-fledged video generation models with temporal priors.
> >
> > Our benchmark is intentionally designed to evaluate a wide range of models, including both T2I- and T2V-based methods, by incorporating tasks that reflect practical editing scenarios. These tasks are critical for identifying the strengths and limitations of different approaches. Thus, this aspect of VEditBench does not represent a shortcoming but rather highlights an area where T2I models can improve, encouraging further advancements in the field.
> >
> >
> > **[Q6: Task Weakness 4 (Multi-Attribute Editing)]**
> >
> > We appreciate the reviewer’s suggestion to incorporate multi-attribute editing into VEditBench. In response, we have expanded the dataset to include this compositional editing task, which involves modifying multiple elements within a scene—such as objects, motion, and style—simultaneously through a single prompt. Further details on multi-attribute editing can be found in Sec. A.4 of the supplementary material.
> >
> > **[Q7: Task Weakness 5 (T2I Limitations on Motion)]**
> >
> > We appreciate the reviewer’s feedback and would like to clarify that this concern highlights a limitation of certain T2I models rather than a shortcoming of our benchmark. While tasks such as motion change and object addition are indeed challenging for T2I models due to their lack of inherent video generative priors, recent advancements—such as temporal adaptations using optical flow—demonstrate that T2I models can achieve competitive temporal consistency.
> >
> > VEditBench is intentionally designed to be model-agnostic, offering standardized tasks that objectively assess the capabilities of both T2I- and T2V-based models. Favoring one type of model over the other would undermine the purpose of a comprehensive benchmark. Instead, VEditBench sets consistent criteria for all models, pushing T2I models to improve their temporal consistency mechanisms while holding T2V models to equally high standards of visual quality. This approach ensures balanced and meaningful evaluation across diverse model architectures.
> >
> >
> > **[Q8: Visualization Weakness (Lack of Qualitative Demos)]**
> >
> > We have included video results in the updated supplementary material. Please refer to Sec. A.3 for more details.

---

### Official Review · Reviewer_edSE · 2024-11-02

**Soundness:** 3
**Presentation:** 4
**Contribution:** 3
**Rating:** 6
**Confidence:** 5

**Summary:**

The authors introduce a text-guided video editing benchmark that includes a large-scale collection of videos across diverse categories, varying durations, and editing tasks grouped into six categories. Additionally, they define nine evaluation metrics to assess outputs across multiple dimensions. By addressing the limitations of previous methods, which relied on private or unreleased datasets and non-standardized evaluation pipelines, this paper seeks to standardize the evaluation of text-guided video editing. The authors also apply their benchmark to evaluate prior approaches, providing a consistent framework for comparison.

**Strengths:**

- I find this benchmark highly valuable for the field, as I agree with the authors' observation that each method currently introduces a unique evaluation pipeline, leading to non-standardized assessments. This work provides a standardized framework for evaluating new methods, which will benefit future research.

- The categorization based on video duration enhances the benchmark, as some methods cannot handle longer videos. This categorization will be useful in differentiating capabilities among approaches.

- The benchmark also covers a wide range of evaluation types. I find the inclusion of motion similarity particularly innovative; if this metric is novel, please cite the relevant sources, as I have not encountered previous works using this measure. Motion similarity offers a meaningful assessment of how well the resulting video maintains motion consistency.

- The paper is very well presented and written.

**Weaknesses:**

- A potential enhancement could be to further categorize object-swapping tasks into two subcategories: one for swapping objects of similar size and another for cases that necessitate substantial motion adjustments (e.g., swapping a car with a bike). These scenarios involve distinct challenges, and such categorization would allow for a more precise assessment of method capabilities.

- Additionally, incorporating GPU requirements and runtime as evaluation metrics would improve the benchmark’s comprehensiveness by enabling comparisons on computational efficiency and scalability.

- I have a question regarding the evaluation of methods with different prompt requirements. Methods such as InstructVid2Vid rely on instructional prompts, while others like TokenFlow and RAVE use target-based prompts. How does the benchmark account for these differences in prompt style? The instructional format used for all editing prompts, as indicated in Supplementary A.1, may not align with some methods that were not trained with instructional prompt styles, potentially impacting their performance.

- In Table 1, it would be beneficial to include the dataset used in RAVE, as it features longer videos categorized by frame count, providing a clear basis for evaluating duration-dependent performance.

- When discussing video duration (e.g., in lines 205 or 207), it would be more informative to also specify the frame rate (FPS), as duration in seconds alone does not provide a complete measure of the content length without this context.

- In line 402, the figure number is missing within the parentheses, which should be corrected for clarity.

**Questions:**

Could you provide qualitative video results for each method in a downloadable format, such as a PDF or a hosted HTML link? Additionally, example prompts from the dataset would be helpful, as they are currently not visible.

Please refer to my questions in the weaknesses section, as addressing these points would allow me to reconsider and potentially increase my score.

---

> ### Author Response · Authors · 2024-11-28
> **Response to Reviewer edSE**
>
> **[Q1: Large/small edit for object swap task]**
>
> Thank you for your insightful suggestion. We agree that distinguishing object-swapping scenarios based on the extent of transformation required can offer a more nuanced evaluation of model performance. Notably, our dataset already incorporates varying levels of edits in the object swap task. These prompts are categorized into two distinct subcategories: large edits, which involve significant shape changes (e.g., swapping a car with a bike), and small edits, where objects are of similar size (e.g., changing a tiger to a lion). We appreciate your feedback and believe this refinement further enhances the benchmark’s comprehensiveness.
> For more details, please refer to Sec. A.5.
>
> **[Q2: GPU requirements and runtime as evaluation metrics]**
>
> Thank you for emphasizing the importance of including GPU requirements and runtime as evaluation metrics. We have integrated these metrics into our benchmark, as detailed in Tab. 2 of the revised paper.
>
> **[Q3: prompt requirements]**
>
> We appreciate the reviewer’s insightful question regarding the potential misalignment between different prompt styles and model training paradigms.
> The design philosophy of VEditBench prioritizes standardization and fairness to ensure a robust evaluation framework. To minimize the potential burden on models, all prompts are constructed with simplicity and explicitness, facilitating clear and accessible instructions.
>
> Models evaluated on the benchmark are expected to perform robustly across various prompt styles, reflecting their ability to adapt to diverse real-world input formats. In rare cases where certain methods are biased towards specific prompt styles, we encourage the use of prompt engineering techniques (e.g., LLMs) as a legitimate and practical approach to aligning prompt styles. These techniques can effectively bridge gaps between a model’s training data and the benchmark’s prompts.
>
> **[Q4: Include the dataset used in RAVE]**
>
> Thank you for the valuable suggestion. In the revised paper, we have updated Tab. 1 to include the dataset information for RAVE.
>
> **[Q5: Specify FPS]**
>
> Thank you for your suggestion. We have clarified the FPS in the revised paper (L205).
>
> **[Q6: Missing figure number in L402]**
>
> Thank you. We have corrected this typo in the revised paper.
>
> **[Q6: Qualitative video results]**
>
> A sample dataset has been included in the updated supplementary material (Sec. A.3). Please visit the project page to explore the video results and example prompts.

---

### Official Review · Reviewer_CJ9A · 2024-11-04

**Soundness:** 3
**Presentation:** 3
**Contribution:** 3
**Rating:** 5
**Confidence:** 3

**Summary:**

This paper proposes VEditBench, a comprehensive benchmark designed for the text-guided video editing (TGVE) task. It contains 420 real-world videos spanning six categories, which are further divided into short (2-4 seconds) and long (10-40 seconds) clips. VEditBench encompasses a wide range of editing tasks, such as object insertion, removal, swapping, etc. It evaluates existing state-of-the-art approaches from multiple dimensions, including semantic fidelity and visual quality, leading to insightful discussions/conclusions in TGVE.

**Strengths:**

- The benchmark for text-guided video editing is both crucial and captivating. The work notably bridges a substantial gap, laying an important foundation for future studies of video editing.
- The paper is crafted with a coherent structure and logical flow, making it accessible and comprehensible.
- The proposed benchmark reveals the capabilities and limits of existing methods, demonstrating the trade-offs in performance across various tasks and dimensions.

**Weaknesses:**

- The absence of visual results makes it difficult to fully assess whether the quantitative metrics align with the intended objectives or functionality.
- [Motion] Motion plays a vital role in video generation and editing, yet the proposed benchmark doesn't fully address this aspect. Is the CLIP score sensitive to the video motion (both object and camera motion) and text description? How to measure the performance of different methods in editing videos with varying levels of motion?
- [Diversity in video] The video collection lacks diversity. While it includes various categories, it's unclear if there is style diversity, such as cartoons and paintings (different levels of abstraction).
- [Diversity in task] Simultaneous multi-element editing with text isn't addressed in VEditBench. Typically, the editing requirements can be combinatorial and complicated.
- [Metric] The structure preservation is not considered, which is common and important. Imagine that users intend to edit the motion of the main character while preserving the overall layout.
- [Metric] The efficiency of different editing methods is not considered in the benchmark.


Typo:
- Lack of figure index in L402.

**Questions:**

- How are the starting points selected for the tracking of motion similarity metric? How significantly does this impact the evaluation?
- How can different methods be measured and compared if they only support editing videos with fixed-length frames (and distinct from each other)?
- Why do most metrics in VEditBench-long show better results compared to VEditBench-short?

---

### Official Review · Reviewer_DUEf · 2024-11-05

**Soundness:** 2
**Presentation:** 3
**Contribution:** 2
**Rating:** 5
**Confidence:** 5

**Summary:**

This paper introduces VEditBench, a new benchmark dataset for text-to-video editing tasks. The benchmark contains 420 real-world videos across six editing tasks. Additionally, the authors propose nine evaluation metrics to assess the semantic fidelity and visual quality of edits. They also provide a quantitative analysis of the performance of ten state-of-the-art text-to-video editing models on the newly introduced benchmark and evaluation metrics.

**Strengths:**

+ The proposed benchmark could be valuable for advancing research in text-to-video editing.
+ The quantitative analysis provided is strong.
+ The paper is generally well-written.

**Weaknesses:**

* The dataset collection and annotation process is not very clear
    - How many videos were collected per category?
    - How was GPT-4o used to caption the videos? What prompts were used? It would be helpful if the authors provided a sample video with corresponding captions obtained through this process.
    - How many people participated in the manual review and revision process?
    - No sample dataset is provided as a reference.
    - Concerns regarding the copyright and continued accessibility of the curated dataset are not addressed.

* The proposed tasks and evaluation metrics are not strongly motivated

    - The proposed editing tasks are not novel, as they have been utilized in prior works. What is the purpose of presenting them as new? Why were these six tasks selected? It is important to discuss how the proposed dataset and tasks differentiate themselves from existing datasets.
    - The inclusion of all nine evaluation metrics seems redundant. Why is each metric necessary? For instance, what is the rationale for a spatio-temporal metric when individual spatial and temporal metrics are already provided?
    - The evaluation process relies heavily on numerous pretrained models, which may introduce errors and impact the reliability of the proposed metrics. Did the authors implement any measures to mitigate potential error propagation?
    - Different editing cases should ideally weigh metrics differently. For instance, spatial alignment is nearly irrelevant for a motion change task unless the caption specifically references motion (which would be rare). This is a critical consideration in designing evaluation metrics and appears to be overlooked in the paper.
    - Did the authors consider using optical flow for the motion similarity metric?


* The experimental analysis in the paper is limited and lacks depth
    - Beyond the qualitative results, what insights can we draw about the models? Why do some models perform well on one metric but poorly on another?
    - Given the limited technical contribution, the submission would benefit significantly from a detailed analysis from the perspectives of model architecture, dataset composition, and training settings.
    - The paper lacks a video analysis. Including a demo video to showcase the different quantitative results would provide a clearer understanding of the outcomes discussed.
    - The quantitative charts do not yield any meaningful insights, as the models seem to behave inconsistently across metrics. To what extent does this issue stem from the evaluation metric design itself?
    - What is the takeaway message of the paper? What are the key research directions that remain unexplored? The paper briefly mentions the need for “specialized architectures and training strategies tailored to the specific challenges of long video editing,” but this statement is vague and lacks depth.

**Questions:**

Please refer to the questions (or issues) mentioned in the Weaknesses section.

---

> ### Author Response · Authors · 2024-11-28
> **Response to Reviewer DUEf (1/3)**
>
> **[Q1: Number of videos per category]**
>
> As described in L211 of Sec. 3.1, we curated a collection of 420 videos, consisting of 300 short videos and 120 long videos, all at a resolution of 720x1280. These videos are evenly distributed across six categories, with each category containing 70 videos (50 short and 20 long).
>
> **[Q2: Prompt generation]**
>
> As outlined in Fig. 2, we employed GPT-4o as part of our data curation pipeline to generate captions by feeding sampled video frames in a grid format alongside video context descriptions. Detailed explanations of the edit prompt generation process are provided in Sec. 3.2 and Sec. A.1. Please refer to Fig. 2 and Fig. 4 for a sample video. See project page in the updated supplementary material for more video examples.
>
> **[Q3: Manual review and revision process]**
>
> We recruited five college students, all native English speakers, to participate in the review and revision process. To ensure high accuracy and minimize errors, each video was independently reviewed by at least three annotators. Discrepancies were resolved through discussion or majority consensus, ensuring the quality of the final dataset.
>
> **[Q4: Sample dataset]**
>
> We have included a sample dataset in the update supplementary material to serve as a reference. Please refer to Sec. A.3 for more details.
>
> **[Q5: Copyright and continued accessibility]**
>
> The videos in VEditBench were carefully curated from publicly available platforms, such as YouTube and Videvo, while strictly adhering to their respective licensing terms. Specifically:
> - License: Videos sourced from YouTube comply with the research-only licensing terms of the Panda-70M dataset. From Videvo, only Creative Commons-licensed videos were selected.
> - Accessibility: VEditBench includes links to legally permissible archived copies to mitigate potential future unavailability. Each video is accompanied by metadata specifying its source and licensing terms for full transparency and traceability.
> - Non-Commercial Use: VEditBench is explicitly designed for non-commercial research. Clear documentation will outline permissible uses and reinforce adherence to the original licensing agreements.
>
> **[Q6: Editing tasks]**
>
> The six editing tasks (*i.e., object addition, object removal, object swap, scene replacement, motion change, and style translation*) were chosen because they represent the most common and practical challenges in real-world video editing. These tasks cover a wide range of use cases, such as adding visual content, removing visual distractions, and  altering motion dynamics.
>
> While previous works, such as LOVEU-TGVE-2023, have included tasks like object swapping, background replacement, and stylization, VEditBench goes further by introducing a wider range of tasks, including object addition, object removal, and motion change—challenges often overlooked in existing benchmarks. Combined with a diverse, large-scale video dataset and a robust multi-dimensional evaluation framework, VEditBench provides a more comprehensive and realistic assessment of model performance, significantly advancing the standard for evaluating video editing capabilities.
>
>
> **[Q7: Evaluation metrics]**
>
> Video editing is indeed multifaceted, requiring the evaluation of spatial appearance, temporal dynamics, and their interactions to ensure coherence. A single metric cannot fully capture this complexity.
>
> While spatial and temporal metrics assess frame quality and motion consistency respectively, they do not account for inconsistencies that arise from their interaction. For example, a video with high spatial quality and smooth motion may still fail to maintain spatial integrity across frames, such as subtle object distortions or shifts. Spatio-temporal metrics provide insights into dynamic scene fidelity that are not captured by separate spatial or temporal evaluations.
>
> By including all nine metrics, VEditBench provides a comprehensive framework to evaluate diverse aspects of video editing, reflecting the nuanced demands of real-world applications.
>
> **[Q8: Reliability of pretrained models]**
>
> We acknowledge the potential concerns regarding reliance on pretrained models. To mitigate error propagation, we carefully selected state-of-the-art pretrained models known for their accuracy and robustness, which have been rigorously validated within their respective domains.
>
> Additionally, we curated a diverse dataset spanning various video categories and editing scenarios to minimize the risk of domain-specific biases or overfitting affecting the evaluation. This ensures that any potential limitations of the pretrained models are balanced by the dataset’s broad coverage.

---

> > ### Author Response · Authors · 2024-11-28
> > **Response to Reviewer DUEf (2/3)**
> >
> > **[Q9: Metric weight]**
> >
> > We acknowledge that the relevance of each metric can vary by editing task. However, even for tasks like motion change, where temporal dynamics are the primary focus, metrics like spatial alignment remain valuable as it measures whether unedited regions in the video maintain their original appearance.
> >
> > While tailoring metric weighting to specific tasks could enhance the evaluation process by aligning with task-specific priorities, it introduces the risk of subjective bias, as individuals may assign different levels of importance to each metric. We plan to explore task-specific metric weighting or composite scoring in future work, leveraging objective criteria or data-driven methods to ensure consistency and fairness while capturing the relative significance of each metric for different tasks.
> >
> > **[Q10: Optical flow for motion similarity]**
> >
> > Optical flow generates dense motion fields, capturing pixel-level movements. While this level of detail is valuable in certain applications, it can make the metric overly sensitive to minor, irrelevant variations, such as noise or subtle differences in motion not perceptible to human viewers. This could lead to misleadingly low similarity scores even for visually coherent edits.
> >
> > Instead, we opted for trajectory-based motion analysis using a robust point-tracking framework. This approach captures key motion patterns without being overwhelmed by pixel-level noise, providing a more balanced and interpretable measure of motion similarity. Please refer to Sec. A.2 for detailed explanation of motion similarity.
> >
> > **[Q11: Insights about models]**
> >
> > Models like DMT and InsV2V, which leverage pretrained text-to-video (T2V) models, generally outperform those adapted from text-to-image models in motion smoothness. This advantage arises from the explicit training of T2V models on video data, which equips them with a motion prior that ensures smoother and more consistent motion dynamics across frames.
> >
> > Early methods such as Tune-A-Video and Text2Video-Zero excel in spatial metrics, producing high-quality individual frames. However, their reliance solely on temporal attention often leads to challenges with temporal coherence, resulting in inconsistencies in motion dynamics. Recent advancements, like TokenFlow and VidToMe, address this limitation by enhancing temporal consistency through techniques such as optical flow and self-attention tokens, all while maintaining strong image quality.
> >
> > **[Q12: Analysis on model architecture, dataset composition, and training settings]**
> >
> > **Model Architecture**: Methods based on text-to-video models (e.g., DMT and InsV2V) often excel in temporal smoothness but tend to sacrifice spatial quality. Conversely, models that rely solely on text-to-image frameworks (e.g., Text2Video-Zero, Vid2Me) demonstrate superior spatial quality. Future research could explore hybrid approaches that combine the strengths of spatial text-to-image models with the temporal coherence of text-to-video models to balance these trade-offs. Additionally, models that incorporate optical flow (e.g., TokenFlow, Flatten) generally achieve better temporal consistency compared to those relying exclusively on temporal attention mechanisms (e.g., Tune-A-Video, Pix2Video).
> >
> > **Dataset Composition**: Methods utilizing models pretrained on large-scale video datasets typically deliver superior temporal smoothness, whereas those trained solely on image datasets often excel in spatial quality.
> >
> > **Training Settings**: Tuning-based methods such as Tune-A-Video and MotionDirector require longer training times and are more susceptible to issues like overfitting and color shifts, which can negatively impact aesthetic quality. In contrast, zero-shot methods like TokenFlow and VidToMe are more generalizable and avoid the common drawbacks of fine-tuning.
> >
> > A more detailed analysis will be provided in the final paper.

---

### Author Response · Authors · 2024-12-04
**Gentle Reminder: Upcoming Paper Discussion Deadline**

As the paper discussion deadline is quickly approaching, we would like to kindly remind reviewers who have not yet had the chance to respond to our rebuttal. We would greatly appreciate it if you could let us know whether our response has sufficiently addressed your concerns or if there are any remaining questions. Should you have any unresolved issues, we would be more than happy to actively address them!

---

### Meta-Review · Area_Chair_on9x · 2024-12-17

**Metareview:**

The paper discusses a new benchmark for the text-guided video editing task. Reviewers acknowledged the importance and the difficulty of the problem. Yet, all of them place the paper around or below the borderline. They listed a number of weaknesses, such as heavy reliance on existing 3rd-party models, lack of experimental analysis, lack of certain types of editing, lack of diversity. They further mention that the used metrics have been previously discussed and analyzed. The AC believes that the problem is a very nuanced and multifaceted, making it hard to be addressed by one single manuscript. Since the paper didn't get the necessary level of support from the reviewers the decision is to reject the manuscript.

**Additional Comments On Reviewer Discussion:**

The reviewers listed a number of questions and issues with the current manuscript. The authors did try to address them, succeeding at some. For example, reviewer DUEf, acknowledged that some of their concerns are addressed. Despite this, they still kept their score below the borderline. It also was not clear if the number of provided additional details is sufficient. The same more-or-less holds for other responses. Essentially each of the reviewers provided a list of 5-10 weaknesses, which authors tried to address, showing that the community has questions about the manuscript, suggesting that the paper requires revising.

---

### Decision · Program_Chairs · 2025-01-22

Reject